# Synchronous High-Grade Squamous Intraepithelial Lesion of the Fimbria of the Fallopian Tube in a 51-Year-Old Woman with Invasive Squamous Cell Carcinoma of the Uterine Cervix

**DOI:** 10.3390/diagnostics13172836

**Published:** 2023-09-01

**Authors:** Anne-Sophie Wegscheider, Nikolas Tauber, Kirsten Graubner, Gudrun Ziegeler, Michael Behr, Christoph Lindner, Axel Niendorf

**Affiliations:** 1MVZ Prof. Dr. Med. A. Niendorf Pathologie Hamburg-West GmbH, 22767 Hamburg, Germany; 2Department of Gynecology, Agaplesion Diakonieklinikum Hamburg, 20259 Hamburg, Germany

**Keywords:** fallopian tube, fimbriae, squamous cell intraepithelial neoplasia, squamous cell carcinoma, carcinoma of uterine cervix, HPV

## Abstract

Primary squamous cell carcinoma or squamous intraepithelial lesion of the fallopian tube is a very rare finding with only a small number of cases worldwide. We describe the case of a 51-year-old woman, undergoing an abdominal hysterectomy after the diagnosis of an HPV-associated invasive squamous cell carcinoma of the uterine cervix with the unexpected detection of an HPV16-positive high-grade squamous intraepithelial lesion of the fimbria of the right fallopian tube in the resection specimen. The finding of an isolated, HPV-associated squamous intraepithelial lesion in the fallopian tube raises the question of a de novo development in this body compartment (after exclusion of a continuous metastatic spread from the uterine cervix) by taking a virus-associated field effect into account and should encourage the inclusion of this possibility when examining the fallopian tube in a routine setting.

Intraepithelial neoplasia of the fallopian tube, by definition of the WHO classification of female genital tumors, includes serous tubal intraepithelial lesions (STIL) or serous tubal intraepithelial carcinoma (STIC). They are regarded as precursor lesions of a high-grade serous carcinoma (HGSC) of the fallopian tube of the ovary, whereas a squamous intraepithelial lesion is not listed in the WHO classification [1]. The epithelium of the fallopian tube can not only undergo neoplastic changes, but can also show variations due to metaplastic changes. Mucinous metaplasia, with additional gastric differentiation, transitional metaplasia (similar to that seen in Walthard cell rests) or squamous metaplasia have been described. Other epithelial alterations include papillary hyperplasia and epithelial pseudoneoplasia associated with salpingitis and Arias-Stella effects. These changes are usually seen following inflammatory processes, and variations in cellular structures depend on reproductive age, hormonal status and location of the tube [2,3,4,5,6]. Generally speaking, a squamous metaplasia can be a precursor for a primary squamous lesion, either intraepithelial or invasive [7]. Nevertheless, squamous lesions in the fallopian tube are rarely described in the literature and the WHO classification does not acknowledge the squamous lesion of the fallopian tube (neither intraepithelial nor invasive). Secondary, metastatic involvement of the fallopian tube in cases with primary carcinoma of the uterine cervix is also not very common and even less so in squamous cell carcinomas than in adenocarcinomas [8,9]. Regarding these cases, the majority do show a continuous infiltrative growth involving the uterine cervix, isthmus and corpus/cavity, spreading in a “carpet-like” pattern [10,11]. But, the evidence of a squamous intraepithelial lesion without local invasion and without infiltration of the uterine corpus is extremely rare, displaying a synchronous and independent lesion rather than a “skip lesion” of the high-grade intraepithelial lesion or the invasive carcinoma of the cervix [12,13].

A 51-year-old woman presented with a positive Pap smear (Pap V, nomenclature München III for gynecological cytodiagnostics; SCC–squamous cell carcinoma, Bethesda system) and a positive test for human papilloma virus (HPV) type 16. Clinical examination revealed a macroscopically visible lesion, leading to a stage FIGO IB (with no evidence of infiltration to the parametria) and a loop electrosurgical excision procedure (LEEP) was performed. A preoperative ultrasound did not show any suspicious changes in the adnexa. The histopathological finding was an HPV-associated, high-grade invasive squamous cell carcinoma with an adjacent high-grade squamous intraepithelial lesion (HSIL) of the uterine cervix. 

Following a frozen section procedure of pelvic sentinel node biopsies (detected with use of indocyanine green, ICG), where no metastatic spread could be detected, a hysterectomy with bilateral adnexectomy (Wertheim-Meigs surgical procedure) was performed. In the postoperative workup of the specimen, the previously diagnosed invasive squamous cell carcinoma showed a diameter of 3.8 cm with an infiltration of the cervix wall of 1.6 cm, the uterine cervix almost completely invaded up to the isthmus without infiltration of the parametria and without angioinvasion of lymph or blood vessels, showing local resection margins without atypia (G3, pT2a1, pN0, L0, V0, R0 according to TNM classification of tumors, 8th edition [14]). Neither the endometrium or myometrium of the uterine corpus nor the isthmus of the tube/intramural part of the uterus showed infiltration by the cervical carcinoma. The histological routine examination of the adnexa unexpectedly revealed several foci of a high-grade squamous intraepithelial lesion with a maximum diameter of 1 mm each within the fimbria of the right fallopian tube (Figure 1A–C).

This diagnosis, which was initially made on the basis of an H&E-stained slide showing abnormal variation in cell shape and size and hyperchromatic nuclei as well as atypical mitotic figures, was complemented with immunohistochemical stains, showing a positivity for p63 and CK5/14 (Figure 2A,B) without an expression of GATA3 (Figure 2D), as a lead for a squamous differentiation, accompanied by positivity for p16 (Figure 2C) and a high proliferation rate (Ki67) without the detection of a p53 mutation.

HPV testing using the APTIMA HPV Assay (HOLOGIC, San Diego CA) on stained smears and stained FFPE tissue sections was internally validated by comparing samples of both types of material (regular Aptima tubes as well as smears/FFPE tissues) available.

There was no evidence of a concomitant serous intraepithelial lesion (STIL or STIC). The left fallopian tube, including the fimbria, and both ovaries did not show neoplastic changes, a peritoneal carcinosis was not detected either.

After presentation of the case to the interdisciplinary tumor board, the patient was advised to undergo vaginal brachytherapy, considering the diagnosis of an invasive carcinoma of the uterine cervix.

To summarize, we could not prove angioinvasion in the primary carcinoma of the uterine cervix, invasive spread throughout the uterine cavity, nor invasion of the parametric tissue or peritoneal carcinosis and there was no evidence of the invasive growth of the only focal lesions in the macroscopically inconspicuous fimbria of the fallopian tube, using H&E staining and immunohistochemical analysis. An intracavitary or continuous expansion therefore seems implausible. So, the question arises, of how this lesion should be interpreted. The most probable explanation would be that an HPV infection over a longer period of time induced additional neoplastic changes in the fallopian tube in analogy to the so-called field effect observed in other tissues or organs such as the uterine cervix, anal canal, vagina, vulva or otolaryngologic area [12,13]. 

In the patient presented here, the diagnosis of the synchronous lesion in the fimbria of the fallopian tube was not clinically or therapeutically relevant at this time, as there was no transition of the HSIL into an invasive carcinoma in the fallopian tube yet. The literature until now gives no information about this biological/clinical behavior. Since such a lesion, as in the case presented here, does not cause any symptoms and is furthermore not detectible by macroscopical inspection, it may be overlooked in patients that are treated for HPV-associated intraepithelial lesions or invasive neoplasia elsewhere in the lower anogenital tract, especially in the uterine cervix. We conclude that squamous lesions should be included in the spectrum of differential diagnosis of neoplastic changes of the fallopian tube.

## Figures and Tables

**Figure 1 diagnostics-13-02836-f001:**
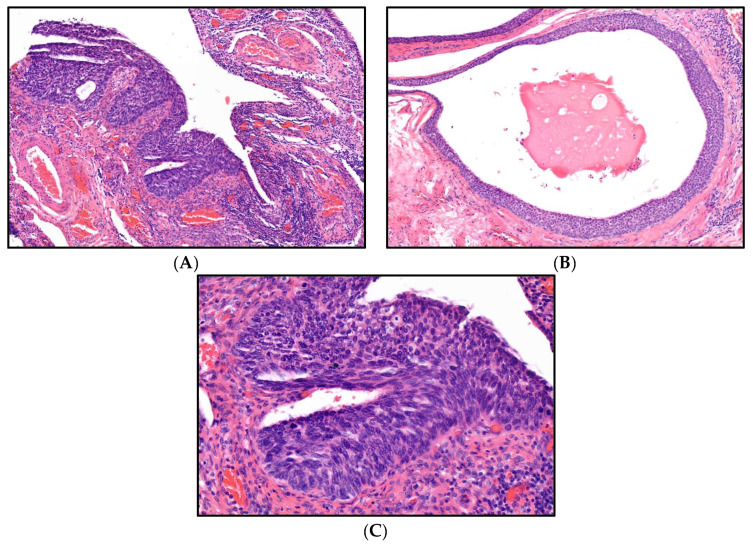
(**A**,**B**): Microscopic sections of two foci of a high-grade squamous intraepithelial lesion (HSIL) in the fimbria of the fallopian tube, overview, H&E stain (magnification level 10×), (**C**): High magnification of HSIL as displayed in (**A**,**B**); H&E stain (magnification level 20×).

**Figure 2 diagnostics-13-02836-f002:**
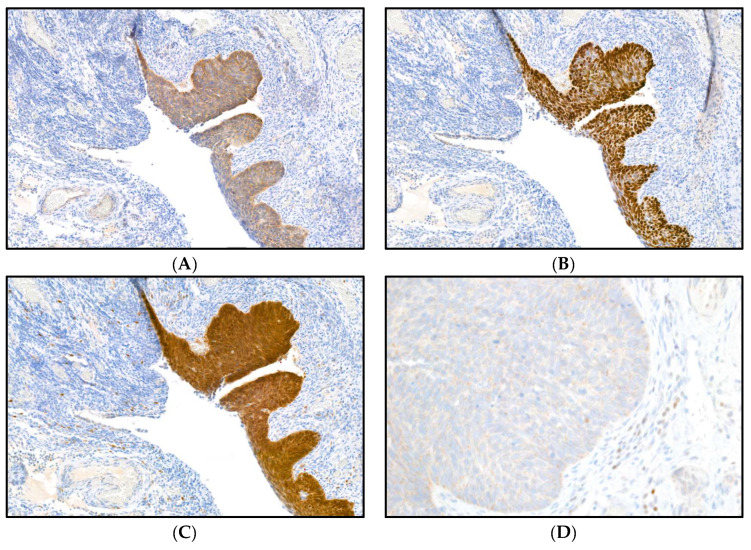
(**A**–**C**): Immunohistochemical staining of the HSIL in the fimbria with CK 5/14 (**A**), p63 (**B**) and p16 (**C**), a potential surrogate marker of persistent high-risk HPV infection [15], which was complemented with a molecular detection of HPV (magnification level 10×), (**D**) GATA3, a nuclear marker, is negative in the HSIL (with positive on-slide control, not shown; magnification level 40×).

## Data Availability

Not applicable.

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
