# Peer review of "Synchronous High-Grade Squamous Intraepithelial Lesion of the Fimbria of the Fallopian Tube in a 51-Year-Old Woman with Invasive Squamous Cell Carcinoma of the Uterine Cervix"

_diagnostics, 2023, doi:10.3390/diagnostics13172836_

Round 1
Reviewer 1 Report
The authors present a case of a woman with invasive cervical squamous cell carcinoma, in whom salpingectomy showed that they interpret as HPV16+ high-grade squamous intraepithelial lesion of the fimbriated end of the right tube. They note this is a rare finding as this is not normally a site with squamous epithelium. This submission is potentially valid, but the results can be presented more clearly. The major issue is that it is not certain that the authors’ conclusion that this is HSIL is correct. The process could be urothelial, since (as they observe) Walthard nests and urothelial metaplasia are rather common in the tube.
MAJOR POINTS
It is unclear why Figure 1 and 2 are separate; they show the same lesion and area. They can be three panels of a single figure.
It is not certain that the tubal lesion is not transitional metaplasia (potentially urothelial carcinoma in situ). The authors report p63 and CK5/14 but these are not shown even though the diagnosis rests upon them. p63 is positive in urothelium and so is CK5. CK14 is said to be a squamous marker but in a cocktail with CK5, it is impossible to know which antibody is reacting. When the diagnosis rests upon details of the IHC, specific clones should be stated. To be honest there is a potential to get any antibody to stain with sufficient adjustment of the protocol. There could be inadvertent positivity. To protect against this, the authors could use known squamous and urothelial tissue as controls.
Figure 3 shows the p16 but it is hard to tell if the area that is staining is the relevant area. The picture should be bigger and taken at higher resolution and/or magnification. In this small picture, it is not clear what is going on. Finding p16+ does not guarantee this is a squamous lesion; urothelial CIS would be positive, https://pubmed.ncbi.nlm.nih.gov/22498670/. Well-known authors proposed a panel of uroplakin III, S-100, GATA3, CK14 and desmoglein to differentiate between urothelial and squamous differentiation (https://pubmed.ncbi.nlm.nih.gov/22995333/).
The authors state that there was molecular detection of HPV in the tube using the Aptima HPV assay. Since this is indeed a molecular assay, not tissue-based, how was the assay done? Does it work on FFPE scrapings?
Seeing RNA in situ hybridization would be more compelling as evidence of an HPV role.
MINOR POINTS
At line 13 (abstract), finding tumor in the tube would not be a coincidental finding or even an incidental one; the purpose of staging is to identify distant disease. It is perhaps “surprising” or “unexpected”.
At lines 13, 15, 65 and elsewhere, please note the Lower Anogenital Squamous Terminology term is “HSIL” (high-grade squamous intraepithelial lesion)—“lesion”, not “neoplasia”.
At line 48, can Pap V in the Munich terminology be crosswalked to U.S. (Bethesda) terminology? The Pap showed invasive squamous cell carcinoma.
Author Response
Dear Sir or Madame,
we want to thank you for your critical and accurate assessment of the manuscript “Synchronous high-grade squamous intraepithelial neoplasia of the fimbria of the fallopian tube in a 51-year-old woman with invasive squamous cell carcinoma of the uterine cervix” for Interesting images.
The diagnosis of a high-grade squamous intraepithelial lesion (HSIL) was confirmed by four pathologists on H&E-slides, considering criteria like hyperchromasia, atypia in cell size and cell size variations combined with atypical mitotic figures in the entire epithelium, e.g. (as distinct from metaplasia). Following your suggestion not to misrender urothelial CIS, we complemented the immunohistochemical tests by GATA3, which was a marker with common positivity in urothelial lesions according to the kindly recommended paper, being negative in the area of the HSIL.
Within the time until the deadline we were not able to add staining of all the markers of the suggested panel, as we do not have all of the antibodies used in daily routine and not all of them are established in our routine laboratory.
HPV-testing using the APTIMA HPV Assay (HOLOGIC, San Diego CA) on stained smears and stained FFPE tissue sections was internal validated by comparing samples of which both types of material (regular Aptima tubes as well as smears/FFPE tissues) were available.
Figures were revised and completed for immunohistochemical slides with higher magnification of the areas of interest.
Minor points were revised, the term high-grade intraepithelial “lesion” was consistently changed and the Bethesda terminology of the Pap smear result was added.
Reviewer 2 Report
It is a very rare case, and although it is an interesting and acceptable article, it needs minor modifications in the following points.
Line 16, de novo should change to an italics figure.
H+E-stain should change to H&E stain.
Line 48, pap smear should change to Pap smear.
Author Response
Dear Sir or Madame,
we want to thank you for the critical assessment of the manuscript “Synchronous high-grade squamous intraepithelial neoplasia of the fimbria of the fallopian tube in a 51-year-old woman with invasive squamous cell carcinoma of the uterine cervix” for Interesting images.
We implemented the minor modifications you suggested in our manuscript, changing the term “H&E-stain” consistently and correcting “Pap smear”.
Reviewer 3 Report
Thank you for submitting the case report. Did this patient have any ultrasonography done earlier so as to comment on the adnexal area?
Can be accepted after answering of the query asked for
Author Response
Dear Sir or Madame,
we want to thank you for the critical assessment of the manuscript “Synchronous high-grade squamous intraepithelial neoplasia of the fimbria of the fallopian tube in a 51-year-old woman with invasive squamous cell carcinoma of the uterine cervix” for Interesting images.
The preoperative examinations of this patient included an ultrasonography, where no suspicious changes of the adnexa could be shown. We added this information in our manuscript.
Round 2
Reviewer 1 Report
The authors have made good-faith revisions along the lines suggested by the reviewers. It is encouraging to hear that GATA3 was negative. I would recommend showing this stain. There is room in Figure 2, which currently has only three panels.
Additional English-language editing could be done to correct minor stylistic lapses.
Author Response
Dear Sir or Madame,
we added the negative GATA3-stain of the HSIL as recommended as an additional panel in Figure 2.
We want to thank you once more for your constructive feedback and advice.